# A Guide to Measuring Heart and Respiratory Rates Based on Off-the-Shelf Photoplethysmographic Hardware and Open-Source Software [note 1]

**DOI:** 10.3390/s24123766

**Published:** 2024-06-10

**Authors:** Guylian Stevens, Luc Hantson, Michiel Larmuseau, Jan R. Heerman, Vincent Siau, Pascal Verdonck

**Affiliations:** 1Department of Electronics and Information Systems—IBiTech, Korneel Heymanslaan, Ghent University, 9000 Ghent, Belgium; pascal.verdonck@ugent.be; 2H3CareSolutions, Henegouwestraat 41, 9000 Ghent, Belgium; info@h3cares.be; 3AZ Maria Middelares Hospital, Buitenring Sint-Denijs 30, 9000 Ghent, Belgium; michiel.larmuseau@mijnziekenhuis.be; 4Partnership of Anesthesia of the AZ Maria Middelares Hospital, Buitenring Sint-Denijs 30, 9000 Ghent, Belgium; jan.heerman@mijnziekenhuis.be; 5Alsico NV, Pont West, 9600 Ronse, Belgium; vincent.siau@alsico.com

**Keywords:** connected care, telemonitoring, vital signs, photoplethysmography

## Abstract

The remote monitoring of vital signs via wearable devices holds significant potential for alleviating the strain on hospital resources and elder-care facilities. Among the various techniques available, photoplethysmography stands out as particularly promising for assessing vital signs such as heart rate, respiratory rate, oxygen saturation, and blood pressure. Despite the efficacy of this method, many commercially available wearables, bearing Conformité Européenne marks and the approval of the Food and Drug Administration, are often integrated within proprietary, closed data ecosystems and are very expensive. In an effort to democratize access to affordable wearable devices, our research endeavored to develop an open-source photoplethysmographic sensor utilizing off-the-shelf hardware and open-source software components. The primary aim of this investigation was to ascertain whether the combination of off-the-shelf hardware components and open-source software yielded vital-sign measurements (specifically heart rate and respiratory rate) comparable to those obtained from more expensive, commercially endorsed medical devices. Conducted as a prospective, single-center study, the research involved the assessment of fifteen participants for three minutes in four distinct positions, supine, seated, standing, and walking in place. The sensor consisted of four PulseSensors measuring photoplethysmographic signals with green light in reflection mode. Subsequent signal processing utilized various open-source Python packages. The heart rate assessment involved the comparison of three distinct methodologies, while the respiratory rate analysis entailed the evaluation of fifteen different algorithmic combinations. For one-minute average heart rates’ determination, the Neurokit process pipeline achieved the best results in a seated position with a Spearman’s coefficient of 0.9 and a mean difference of 0.59 BPM. For the respiratory rate, the combined utilization of Neurokit and Charlton algorithms yielded the most favorable outcomes with a Spearman’s coefficient of 0.82 and a mean difference of 1.90 BrPM. This research found that off-the-shelf components are able to produce comparable results for heart and respiratory rates to those of commercial and approved medical wearables.

## 1. Introduction

Trends like the growing geriatric population (projected to increase by 50% by 2050) [1], the escalating prevalence of chronic diseases [2] and the surge in patient-centric approaches [3] are putting pressure on hospital capacities and healthcare systems around the world [4]. Wearables and innovative technology are a rising field of interest in monitoring health and well-being [5,6,7]. They hold the potential to reduce the frequency of (re)hospitalizations and enhance patient satisfaction [8,9,10]. Among the most widely embraced consumer devices for monitoring purposes are smartwatches, including Fitbits, Apple Watches, and other similar products. These devices track various parameters such as heart rate (HR), sleep patterns, and activity levels, providing users with insights into their overall well-being and lifestyle [11]. The Google Fitbit and the Apple Watch have effectively bridged the gap between lifestyle gadgets and medical instruments, offering features that integrate consumer appeal with clinical applicability. Notably, select functionalities of these wearables have garnered approval from the Food and Drug Administration (FDA), marking a significant endorsement of their medical utility [12]. The predominant technology employed in these wrist-worn devices is photoplethysmography (PPG) [13,14]. A photoplethysmography (PPG) sensor comprises a light-emitting diode (LED) and a photodetector (PD). This sensor operates using two primary modes: transmission or reflection. In transmission mode, light is emitted through the tissue and detected on the other side, while in reflection mode, light is emitted and detected on the same side of the tissue. The PPG technique leverages the interaction of light with biological tissues, such as bone, skin pigment, and both venous and arterial blood. Specifically, blood exhibits stronger light absorption compared to surrounding tissues. The pulsatile component of arterial blood leads to variations in light absorption and reflection, which are detected via the photodetector. These variations in light intensity provide information about blood volume changes in the tissue, allowing for the measurement of physiological parameters such as heart rate and oxygen saturation [15]. PPG devices can detect changes in blood flow as alterations in the intensity of reflected light [14,16]. Parameters including heart rate [17], respiratory rate (RR) [18,19,20,21], oxygen saturation [22,23,24], and even blood pressure (BP) [25] can be derived from the PPG signal. The quality of the PPG signal typically hinges on several factors, including the selection of hardware components (such as LED and PD), their configuration (e.g., quantity, positioning, and utilized wavelengths), and the presence of artifacts (e.g., motion and ambient light) [26]. Although various FDA-approved and Conformité Europeénne (CE)-marked devices are commercially available, they are often expensive and do not facilitate the export of data, thereby posing challenges for use in clinical studies [27].

## 2. Methodology

**Ethical approval declarations.** This study was performed at General Hospital Maria Middelares in Ghent, Belgium. The ethics committee of General Hospital Maria Middelares approved the study (permission number: MMS.2023.048). Written, informed consent was obtained from all subjects included in the study. All procedures performed were in accordance with the 1964 Declaration of Helsinki and its later amendments or comparable ethical standards.

### 2.1. Study Goal and Design Criteria

The primary aim of this study was to examine whether the integration of off-the-shelf hardware components and open-source software could yield accurate vital sign measurements, specifically HR and RR, and to compare these measurements with those obtained from more expensive commercially available medical devices endorsed through CE marking and FDA approval. The aim was to develop an off-the-shelf, easily implementable, and open-source device capable of providing users access to PPG signals for further research or clinical studies. In establishing the design criteria, key factors such as wavelength, voltage requirements, physical dimensions, and cost considerations were taken into account. The research of Maeda et al. [28] concluded that, compared to the traditional red light, green light has better resilience for motion artifacts. Additionally, Lee et al. [29] reported a higher absorption rate of hemoglobin with green light. Consequently, LEDs with wavelengths ranging between 495 nm and 570 nm were selected. When selecting software packages to calculate heart rate (HR) and respiratory rate (RR), the criteria included the availability of a freely accessible source code, ease of implementation without the need for extensive modifications, and the utilization of a coding language that is straightforward to interpret.

### 2.2. Hardware Components

Four break-out chips were found to be commercially available online and subjected to evaluation compared to predefined requirements (see Section 2.1). Based on the results in Table 1, the PulseSensor (https://pulsesensor.com/ (accessed on 13 February 2024)), featuring a peak sensitivity of 565 nm (Avago APDS-9008 photodiode) was determined as the optimal choice. This sensor includes a universal Low-Pass Filter (LPF) employing a passive RC configuration with a resistor (R) value of 100 and a capacitor (C) value of 4.7 uF. For the microprocessor, the Arduino Nano 33 IoT was selected due to its compact form factor, affordability, and inherent support for Bluetooth Low-Energy (BLE) and Wi-Fi connections. To power both the Arduino and PulseSensors, a lithium-ion battery with an output voltage of 3.7 V and a capacity of 820 mAh was chosen, given its widespread availability and cost-effectiveness. The TP4056 5V 1A Lithium Li Micro USB Charging Controller was incorporated between the battery and the Arduino to regulate the charging and discharging processes. A picture of the final prototype utilized in the study is shown in Figure 1.

### 2.3. Software Components

In evaluating software components specific to photoplethysmography (PPG), several open-source packages were considered. For heart rate (HR) calculations, the following packages were systematically assessed: SciPy [30], HeartPy, which employs the methodology of Van Gent [31], and Neurokit [32], which integrates the approach proposed by Elgendi et al. [33] into its processing pipeline. Regarding RR, different methods based on HR were adopted from open-source packages. These included VanGent2019 [19], Soni2019 [20], Charlton2016 [18], Sarkar2015 [21], and Vallat2018 [34]. These five algorithms employed for RR calculation can be categorized based on their underlying methodologies. Sarkar [21], Vallat2018 [34] and Van Gent [31] use ECG-Derived Respiration (EDR), which is acquired through heart rate variability (HRV). Soni [20] calculates RR based on Peak Amplitude Variation (PAV), wherein variations in the baseline indicative of respiratory-induced intensity fluctuations (RIIV) are identified by connecting minima points of the PPG. Charlton’s [18] algorithm integrates three approaches: baseline wander (BW), amplitude modulation (AM), and frequency modulation (FM). This algorithm emerged as the preferred choice in a review conducted by Charlton et al. [18], which evaluated 314 different combinations. Our evaluation involved comparing all possible cross-combinations (3 × HR and 5 × RR) of these various algorithms.

### 2.4. Off-the-Shelf Prototype

Four individual PulseSensors were integrated into a custom-designed bracelet, with the textile component of the bracelet designed by Alsico NV. Each sensor was connected to an Arduino Nano 33 IoT microcontroller to capture and transmit signals via Bluetooth low energy (BLE). The connections were established in accordance with the manufacturers’ guidelines, with the PulseSensors linked to analog input pins of the Arduino (A0 to A3). The power and ground cables from the four sensors were soldered onto a copper strip and connected to the 3.3 V and ground pins of the Arduino, respectively. A rechargeable battery was then connected to the Arduino, with the positive terminal connected to *V*_*in* and the negative terminal to the ground via the charging controller. Figure 2 represents schematics detailing the configuration of the off-the-shelf PPG device.

Based on the literature [26,35,36,37,38,39], different LED and PD configurations have been assessed. Among these, the narrow-line-bottom configuration yielded the most favorable outcomes in terms of signal amplitude and the signal-to-noise ratio. In accordance with this configuration, the PulseSensor LEDs and PDs were positioned on the wrist, specifically targeting the radial and ulnar arteries, as indicated in Figure 3.

### 2.5. Experimental Setup

Figure 4 represents the experimental setup employed for the measurements. As a CE-marked and FDA-approved device, the Vivalink Cardiac Patch was chosen. This device is certified for measuring HR and RR based on a single lead ECG signal. The HR and RR were calculated via the Vivalink software development kit (SDK). The Cardiac Patch was affixed to the right chest in accordance with the manufacturer’s instructions by a specific study nurse, while the off-the-shelf PPG device was secured around the dominant wrist. In the experimental set-up, the Vivalink Cardiac Patch and the off-the-shelf PPG device were connected via BLE to a personal computer to facilitate continuous data capturing.

The off-the-shelf PPG-device was evaluated using a cohort of 15 healthy volunteers. The inclusion criteria comprised healthy volunteers aged 18 years or older who provided signed, informed consent. The exclusion criteria comprised pregnant women, children, individuals with cardiac ailments, allergies, skin conditions, or those lacking decision-making capacity. Prior to commencing the experiment, participant characteristic data, including age, gender, height, wrist circumference, and skin type [40], were recorded (see Table 2). Subsequently, the PPG device was applied to the volunteer’s wrist, and the Vivalink Cardiac Patch was positioned on the chest. Volunteers were instructed to remain motionless and relax for five minutes to acclimate to the room environment before initiating the measurement procedure. PPG signals, HR, and RR were recorded in four distinct positions: supine, seated, standing, and walking in place. Each position was maintained for a duration of three minutes.

### 2.6. Post-Processing of the Raw Signals

Signals obtained from the off-the-shelf PPG device are depicted for a duration of approximately ±10 s in Figure 5. Within the signal, distinct peaks corresponding to blood pulses are clearly visible.

Initially, three distinct photoplethysmographic (PPG) peak-finding algorithms (implemented in Scipy, Neurokit, and HeartPy) were employed. Subsequently, a moving average filter with a ten-second window was applied to the signal to mitigate random artifacts in the calculated heart rate (HR). Following this, HR was determined based on the identified peaks for the Scipy and HeartPy algorithms, as these algorithms exclusively identified peaks. Conversely, the Neurokit package promptly computed the HR within its processing pipeline. Upon a closer examination of this pipeline, it was observed that the exact same formula was utilized for HR calculation across all methods. HR was calculated using Formula (Equation 1) [41].
(1)HR=60/(R−R(s))

Fourthly, for each calculated HR, five distinct methods for estimating RR based on HR were employed. Subsequently, by-the-minute averages of both HR and RR were computed.

### 2.7. Statistical Analysis

The average calculated HR and RR of each measured minute (1, 2, and 3) were compared with the averaged HR and RR of the Vivalink Cardiac ECG patch using Bland–Altman analysis [42]. Additionally, the correlation between the two measurement methods was assessed using Spearman’s coefficient [43].

## 3. Results

### 3.1. Supine Position

Based on the mean differences observed in the Blant–Altman analysis and Spearman’s coefficient in the correlation analysis, the Neurokit algorithm emerged as the preferred choice for HR estimation. For HR, the Bland–Altman analysis revealed a mean difference of 2.80 BPM and a confidence interval of [−10.21–15.80] BPM. Considering an acceptable deviation of less than ±5% for HR, the method falls within the acceptable range, based on criteria from the literature [44,45,46,47,48]. Utilizing the criteria established by Hinkle et al. [49], Spearman’s correlation demonstrated a very high positive correlation (ρ = 0.83) between the Neurokit algorithm and the Vivalink cardiac patch. Additionally, the Scipy algorithm exhibited a high positive correlation as well. For RR, the combination of the Heartpy and Sarkar algorithms was identified as the optimal choice for the supine position. The Bland–Altman analysis resulted in a mean difference of 0.37BrPM and a confidence interval of [−4.93–5.67] BrPM for RR. Considering an acceptable deviation of less than ±10% for respiratory rate, the method also falls within the acceptable range, based on criteria from the literature [44,45,46,47,48]. Spearman’s correlation showed a high positive correlation (ρ = 0.68) between the Heartpy–Sarkar combination and the Vivalink cardiac patch (see Table 3).

### 3.2. Seated Position

In the seated position, the highest correlations for HR and RR were observed. For HR, Bland–Altman analysis revealed a mean difference of 0.59 beats per minute (BPM) and a confidence interval of [−8.64, 9.81] BPM for the most accurate algorithm, Neurokit. Given that an acceptable deviation for heart rate is less than ±5%, the method falls within the acceptable range. Employing the criteria of Hinkle et al. [49] revealed that the best algorithm (Neurokit) has a very high positive correlation (ρ = 0.9) with the Vivalink cardiac patch (refer to Figure 6 and Figure 7). In terms of RR, Spearman’s correlation revealed a very high positive correlation (ρ = 0.82) between the best algorithm (Charlton) and the Vivalink cardiac patch. The Bland–Altman analysis resulted in a mean difference of 1.90 BrPM and a confidence interval of [−1.47−5.26] BrPM for the Neurokit–Charlton combination (see Figure 8 and Figure 9). Given an acceptable deviation of less than ±10% for respiratory rate, the method falls within the acceptable range. The results are summarized in Table 4.

### 3.3. Standing Position

In the standing position, the Bland–Altman analysis resulted in a mean difference of 1.10 BPM and a confidence interval of [−18.97–21.18] BrPM for the Neurokit algorithm, leading to a larger variation between participants. Given an acceptable deviation of less than ±5% for heart rate, the method falls within the acceptable range. According to the criteria of Hinkle et al. [49] the best (Neurokit) algorithm has a high positive correlation (ρ = 0.77) with the Vivalink cardiac patch. For RR, Spearman’s correlation shows a moderate positive correlation (ρ = 0.47) between the Neurokit–Charlton combination with the Vivalink cardiac patch. The Bland–Altman analysis resulted in a mean difference of 1.90 BrPM and a confidence interval of [−1.47–5.26] BrPM for this Neurokit–Charlton combination. Given an acceptable deviation of less than ±10% for respiratory rate, the method falls within the acceptable range. The results are summarized in Table 5.

### 3.4. Walking Position

Walking yielded the poorest results, with the highest occurrence of outliers and divergent values attributed to motion artifacts in the PPG signal. The Bland–Altman analysis indicated a mean difference of −1.58 BPM and a confidence interval of [−23.92–20.76] BPM for the best (Neurokit) algorithm, leading to an even larger variation between participants. Given an acceptable deviation of less than ±5% for heart rate, the method falls within this acceptable range. However, according to the criteria of Hinkle et al. [49], the Neurokit algoritm has a low positive correlation (ρ = 0.29) with the Vivalink cardiac patch. Regarding RR, Spearman’s correlation indicates a low positive correlation (ρ = 0.31) between the Neurokit–Soni combination and the Vivalink cardiac patch. The Bland–Altman analysis resulted in a mean difference of 4.68 BrPM and a confidence interval of [−1.35–10.71] BrPM for this Neurokit–Soni combination. Given an acceptable deviation of less than ±10% for respiratory rate, the method falls outside the acceptable range. The results are summarized in Table 6.

## 4. Discussion

Photopletysmograpic signals are highly susceptible to motion artifacts and noise [50]. These artifacts are influenced by participants’ behavior. Despite instructions to remain still, subtle movements could not be eliminated. The sitting and supine positions exhibited fewer movements, resulting in fewer motion artifacts compared to the standing and walking positions. Although some sources in the literature indicate that sweating decreases the accuracy of PPG signals [51], the literature is ambiguous [52]. Consequently, exercise was not included in our study design. Noise artifacts may be attributed to the construction of the off-the-shelf sensor, which was of a DIY nature. Additionally, long wire connections in the prototype contributed to the noise present in the captured signals [53]. Improvements in the wire-to-chip connections, shortening the wire connections, and precisely soldering points are recommended to enhance the signal quality [54]. Future research may involve integrating all hardware components onto one flexible, integrated chip to directly measure and process all signals.

This study evaluated only three open-source Python software packages, though the literature offers a variety of algorithms [18]. The selected packages aimed to demonstrate the potential of open-source software for HR and RR measurements. Among them, HeartPy [31] only achieved a 50% success rate in calculating HR due to its strict thresholds regarding peak rejections. Thresholds are determined as $RR_{i}nterval_{mean}$ ±30% of $RR_{i}nterval_{mean}$, with maximum and minimum values of + or −300, for the upper and lower thresholds, respectively. If the RR interval exceeds one of the thresholds, it is ignored. The Scipy [30] algorithm uses the same underlying method [31] as the HeartPy algorithm but without the strict thresholds. The superior performance of the Neurokit [32] algorithm is attributed to its comprehensive processing pipeline, including signal cleaning and filtering. Future research could extend these algorithms with Matlab packages, more advanced open-source neural networks, or deep learning algorithms [55,56], albeit requiring a larger participant population. A large amount of possible cleaning and filtering methods are available in the literature [57,58,59,60], but these were not included in the scope of this study and could be subject of future research. The fact is that none of the tested algorithms led to consistent results. This implies that a general approach is less suitable, and a more device-specific approach is needed.

Despite strong evidence that abnormalities in RR are an early predictor of patient deterioration [61,62,63], recording inaccuracies in RR measurement are very frequent [64,65,66]. Lovett et al. [67] found that nurses’ measurements of RR and electronic measurements of RR showed low sensitivity in detecting bradypnea and tachypnea. In their Bland–Altman analysis, nurses’ measurements and electronic measurements of RR showed poor agreement with standard measurement methods like manual counting or capnography. The difficulties in measuring RR are a hurdle that many manufacturers and researchers face. Based on the literature [61,68], expertise and cooperation with different intensive care units in hospitals, and the practical implementations of monitoring devices and telemonitoring critically ill patients, the key message is that trends in vital signs are more important than absolute values and momentary measurements. The mean difference of 2 BrPM in our best combination algorithm remained high, but Spearman’s correlation of 0.82 shows that the device is capable of correlating with the medical-grade device and able to represent trends in respiration measurements.

## 5. Conclusions

An off-the-shelf sensor was created to measure photoplethysmograpic signals and calculate heart and respiratory rates. Based on the design criteria, a configuration of four PulseSensors in a narrow horizontal line at the bottom of the wrist was used as an optical sensor. Data from the PulseSensors were collected with an Arduino nano 33 IOT. Three different methods available in open-source Python packages for heart rate calculation were compared in four different positions: lying, sitting, standing, and walking. After calculating heart rates, five different methods were compared to calculate respiratory rates. Out of the three methods to calculate heart rates, the Neurokit process pipeline performed best overall. For respiratory rates, Neurokit–Sarkar showed a high positive correlation in the seated position. In the standing position, a good correlation was only achieved for heart rate. In the walking position, no high correlations were found for heart rate or respiratory rate. The fact is that none of the tested algorithms led to consistent, good results, and a more device-specific approach is needed. A DIY guide to designing a sensor in order to measure heart and respiratory rates based on off-the-shelf hardware and open-source photoplethysmographic software is now available.

## Figures and Tables

**Figure 1 sensors-24-03766-f001:**
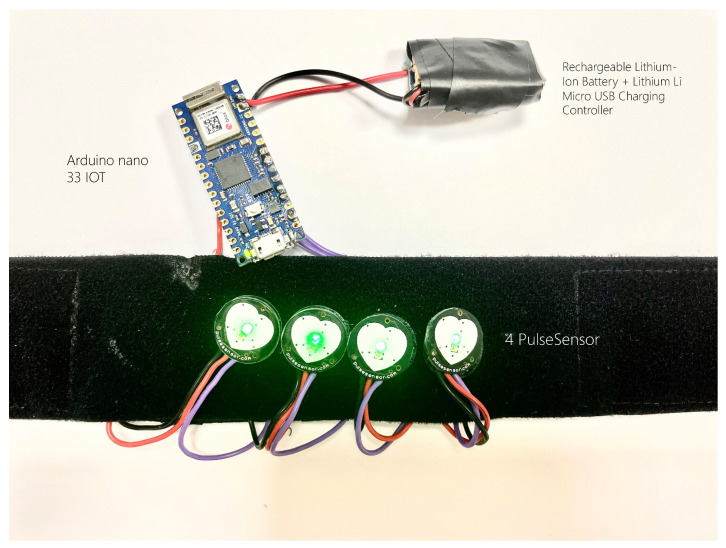
Image of the final prototype of the created off-the-shelf PPG device containing four PulseSensors, one Arduino Nano 33 IoT, a charging controller, and one rechargeable battery.

**Figure 2 sensors-24-03766-f002:**
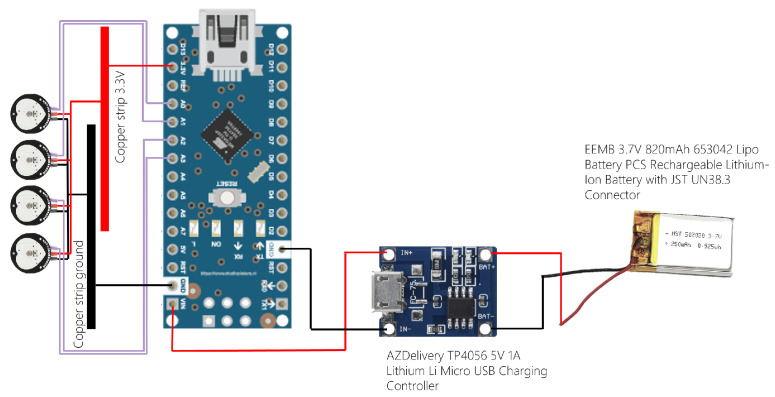
Schematics detailing the configuration of the off-the-shelf PPG device. Four PulseSensors, one Arduino Nano 33 IoT, a charging controller, one rechargeable battery, and their connections are indicated.

**Figure 3 sensors-24-03766-f003:**
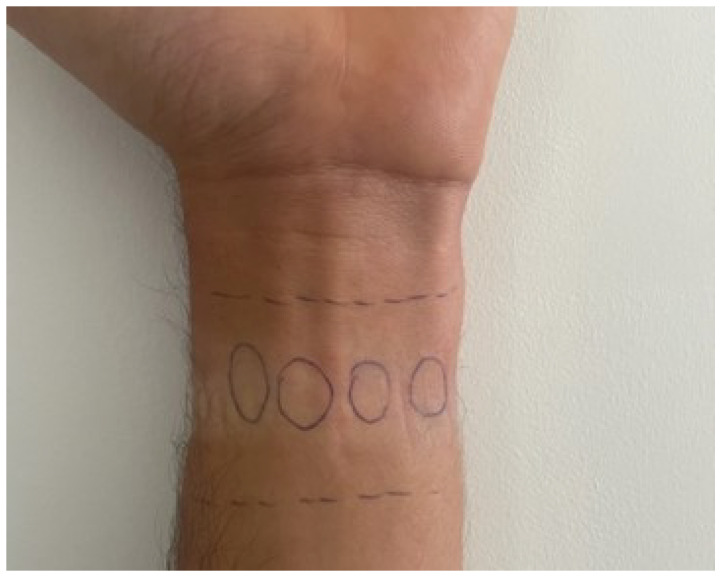
The narrow-line-bottom configuration was chosen as the final configuration, and the PulseSensor LEDs and PDs were positioned on the wrist, targeting the radial and ulnar arteries.

**Figure 4 sensors-24-03766-f004:**
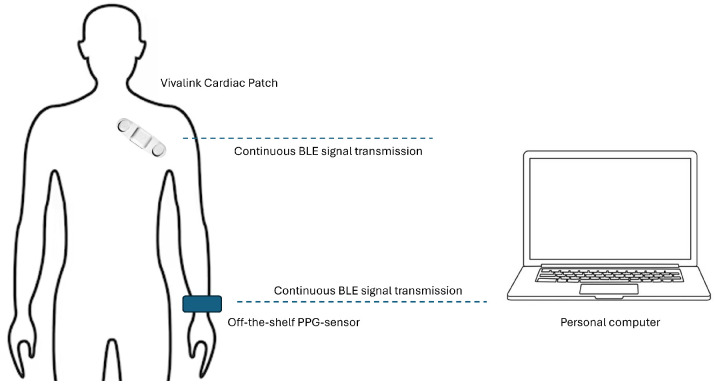
Experimental setup for the PPG measurements with the Vivalink Cardiac patch and the off-the-shelf PPG device.

**Figure 5 sensors-24-03766-f005:**
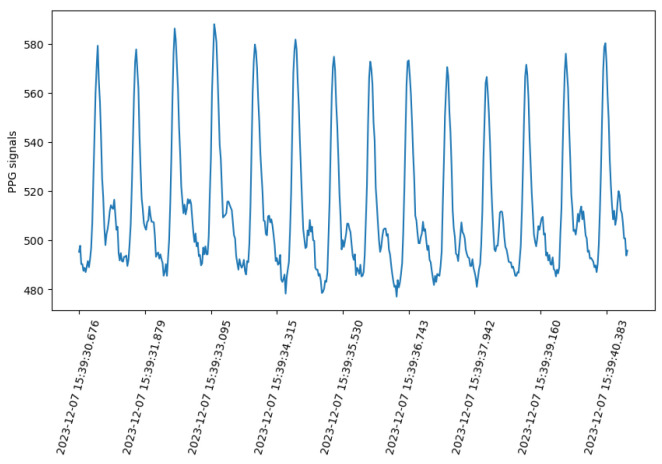
The raw PPG signal from the off-the-shelf PPG device of one of the participants during a 10 s interval.

**Figure 6 sensors-24-03766-f006:**
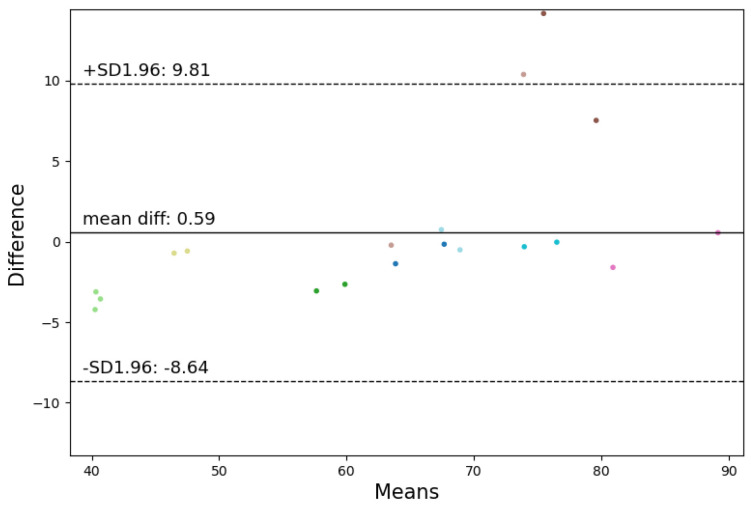
Bland–Altman analysis of the Neurokit algorithm for heart rate calculation in the seated position, revealing a mean difference of 0.59 BPM and a confidence interval of [−8.64–9.81] BPM.

**Figure 7 sensors-24-03766-f007:**
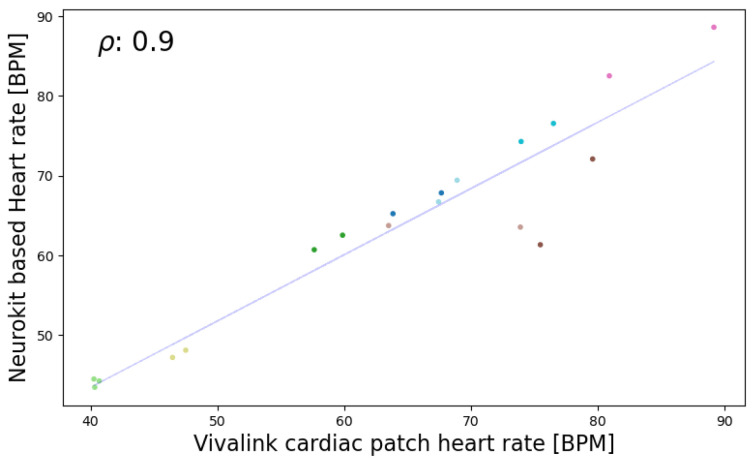
Spearman’s correlation analysis (ρ = 0.9) of the Neurokit algorithm for heart rate calculation in the seated position.

**Figure 8 sensors-24-03766-f008:**
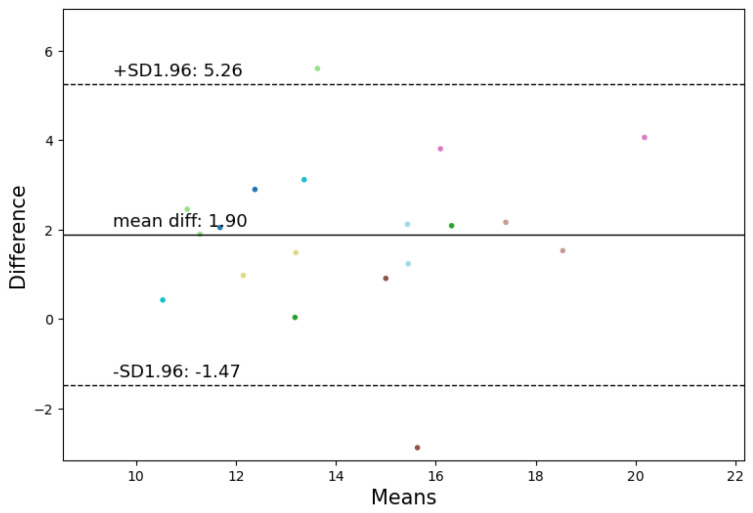
Bland–Altman analysis of the Neurokit and Charlton combination for respiratory rate calculation in the seated position, revealing a mean difference of 1.90 BrPM and a confidence interval of [−1.47–5.26] BrPM.

**Figure 9 sensors-24-03766-f009:**
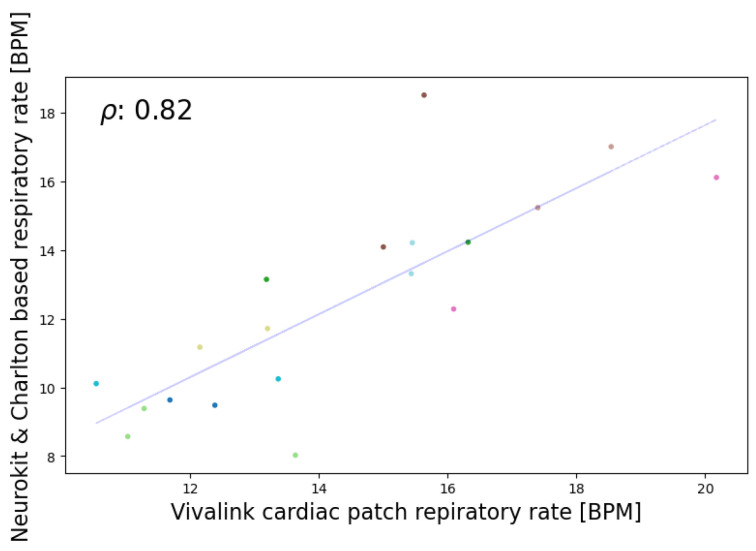
Spearman’s correlation analysis (ρ = 0.82) of the Neurokit and charlton combination for respiratory rate calculation in the seated position.

**Table 1 sensors-24-03766-t001:** Four commercially available, online PPG-sensor break-out chips were evaluated according to their wavelength, required voltage, dimensions, and pricing. The PulseSensor was determined as the optimal choice based on price and wavelength.

Requirements	GY-MAX30102	PulseSensor	MIKROE-4986	MAX30101, MAX32664 (Qwiic)	ProtoCentral AFE4490
Raw data	Yes	Yes	Yes	No	Yes
Light	IR & Red	Green	Red	IR and Red	Red
Dimensions	12.7 × 12.7 mm	ø16 mm	42.9 × 25.4 mm	25.4 × 12.7 mm	55 × 34 mm
Voltage	3.3 V	3.3 or 5 V	3.3 or 5 V	3.3 or 5 V	3.3 or 5 V
Price	€2.95	€24.95	€40.87	€42.29	€73.65

**Table 2 sensors-24-03766-t002:** Population overview of a cohort of 15 healthy volunteers. Age, gender, length, wrist circumference, and skin type were noted as characteristic data points.

Parameters	Averages ± Std
Age	30.26 y ± 5.23 y
Gender	6 (M), 9 (F)
Length	174.33 cm ± 8.8 cm
Wrist circumference	17.6 cm ± 0.9 cm
Skin type	5 (II), 7 (III), 1 (IV), 2 (V)

**Table 3 sensors-24-03766-t003:** Results from Bland–Altman analysis and Spearman’s correlation in the supine position. The Neurokit algorithm (ρ = 0.83) emerged as the preferred choice for HR. For RR, the combination of the Heartpy and Sarkar algorithms (ρ = 0.68) was identified as the optimal choice.

Supine Position			
Methods	Mean Diff.	95% Confidence Interval	*ρ*
**HR**	[BPM]	[BPM]	
Scipy	5.73	−17.31–28.77	0.71
**Neurokit**	2.80	−10.21–15.80	**0.83**
HeartPy *	−37.21	−235.56–161.15	0.58
**RR**	[BrPM]	[BrPM]	
Scipy			
Van Gent	2.45	−2.99–7.88	0.17
Soni	2.50	−3.70–8.69	0.19
Charlton	0.15	−6.46–6.77	0.48
Sarkar	−0.86	−6.98–5.26	0.41
Vallat	−2.14	−13.81–9.53	0.51
Neurokit			
Van Gent	3.36	−2.57–9.30	0.24
Soni	4.51	−1.79–10.81	0.23
Charlton	2.80	−2.78–8.37	0.40
Sarkar	2.25	−2.96–7.46	0.36
Vallat	4.03	−3.12–11.18	0.36
HeartPy *			
Van Gent	3.47	−2.66–9.59	0.22
Soni	4.21	−1.72–10.13	0.32
Charlton	1.50	−4.90–7.90	0.64
**Sarkar**	**0.37**	**−4.93−5.67**	**0.68**
Vallat	2.90	−15.83–10.03	0.75

Best algorithms are indicated in bold. * HeartPy HR and RR could only be calculated for 7/15 participants.

**Table 4 sensors-24-03766-t004:** Results from Bland–Altman analysis and Spearman’s correlation in the seated position. The Neurokit algorithm (ρ = 0.9) emerged as the preferred choice for HR. For RR, the combination of the Neurokit–Charlton algorithm (ρ = 0.82) was identified as the optimal choice.

Seated Position			
Methods	Mean Diff.	95% Confidence Interval	*ρ*
**HR**	[BPM]	[BPM]	
Scipy	5.23	−17.45–27.92	0.55
**Neurokit**	0.59	−8.64–9.81	**0.9**
HeartPy *	−3.65	−46.46–39.15	0.56
**RR**	[BrPM]	[BrPM]	
Scipy			
Van Gent	1.82	−1.52–5.16	0.68
Soni	1.81	−3.35–6.97	0.45
Charlton	0.83	−4.20–5.85	0.61
Sarkar	−0.04	−4.24–4.16	0.67
Vallat	−0.96	−8.27–6.35	0.32
Neurokit			
Van Gent	2.31	−1.81–6.42	0.66
Soni	3.02	−1.10–7.14	0.68
**Charlton**	1.90	−1.47–5.26	**0.82**
Sarkar	1.67	−1.85–5.18	0.79
Vallat	2.88	−1.29–7.04	0.75
HeartPy *			
Van Gent	0.79	−8.04–9.62	0.42
Soni	−0.31	−12.46–11.84	−0.1
Charlton	−1.37	−13.90–11.16	−0.22
Sarkar	−1.62	−12.42–9.18	−0.1
Vallat	−1.35	−8.56–5.85	0.45

Best algorithms are indicated in bold. * HeartPy HR and RR could only be calculated for 7/15 participants.

**Table 5 sensors-24-03766-t005:** Results from Bland–Altman analysis and Spearman’s correlation in the standing position. The Neurokit algorithm (ρ = 0.77) emerged as the preferred choice for HR. For RR, the combination of the Neurokit–Charlton algorithms (ρ = 0.47) was identified as the optimal choice.

Standing Position			
Methods	Mean Diff.	95% Confidence Interval	*ρ*
**HR**	[BPM]	[BPM]	
Scipy	11.19	−17.54–39.91	0.51
**Neurokit**	1.10	−18.97–21.18	**0.77**
HeartPy *	−15.29	−114.71–84.14	0.29
**RR**	[BrPM]	[BrPM]	
Scipy			
Van Gent	3.29	−2.28–8.85	0.22
Soni	1.97	−5.05–8.99	0.14
Charlton	−1.18	−8.68–6.32	0.39
Sarkar	−1.37	−8.89–6.16	0.32
Vallat	−5.61	−18.14–6.92	0.31
Neurokit			
Van Gent	4.18	−1.77–10.14	0.08
Soni	4.68	−1.35–10.71	0.30
**Charlton**	**2.25**	−4.39–8.88	**0.47**
Sarkar	1.68	−5.31–8.67	0.29
Vallat	4.76	−3.55–13.07	0.33
HeartPy *			
Van Gent	5.64	−0.92–12.20	0.04
Soni	5.13	−3.84–14.11	−0.09
Charlton	0.54	−12.09–13.17	0.22
Sarkar	1.10	−10.80–13.01	0.26
Vallat	−5.15	−18.52–8.22	0.30

Best algorithms are indicated in bold. * HeartPy HR and RR could only be calculated for 6/15 participants.

**Table 6 sensors-24-03766-t006:** Results from Bland–Altman analysis and Spearman’s correlation in the walking position. The Neurokit algorithm (ρ = 0.29) emerged as the preferred choice for HR. For RR, the combination of the Neurokit–Soni algorithm (ρ = 0.31) was identified as the optimal choice.

Walking Position			
Methods	Mean Diff.	95% Confidence Interval	*ρ*
**HR**	[BPM]	[BPM]	
Scipy	30.83	−10.73–72.39	−0.47
**Neurokit**	−1.58	−23.92–20.76	**0.29**
HeartPy *	−42.15	−265.18–180.87	−0.53
**RR**	[BrPM]	[BrPM]	
Scipy			
Van Gent	4.58	−6.55–15.72	−0.16
Soni	5.12	−5.50–15.75	−0.05
Charlton	4.77	−6.15–15.69	−0.05
Sarkar	3.89	−7.15–14.92	−0.18
Vallat	1.72	−11.52–14.96	−0.07
Neurokit			
Van Gent	4.18	−1.77–10.14	0.23
**Soni**	4.68	−1.35–10.71	**0.31**
Charlton	2.25	−4.39–8.88	0.2
Sarkar	1.68	−5.31–8.67	0
HeartPy *			
Van Gent	4.52	−6.01–15.05	0.04
Soni	4.07	−6.26–14.40	0.17
Charlton	−1.10	−12.82–10.62	0.29
Sarkar	−0.14	−12.12–11.83	0.18
Vallat	1.41	−11.64–14.47	0.02

Best algorithms are indicated in bold. * HeartPy HR and RR could only be calculated for 10/15 participants.

## Data Availability

The data that support the findings of this study are available on request from the corresponding author, G.S.

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
