# Peer review of "A Guide to Measuring Heart and Respiratory Rates Based on Off-the-Shelf Photoplethysmographic Hardware and Open-Source Softwareâ€"

_sensors, 2024, doi:10.3390/s24123766_

Round 1
Reviewer 1 Report
Comments and Suggestions for Authors
In this manuscript, a guide to measure heart- and respiratory rate is presented. The study is interest to readers.
However, there are several small problems in the manuscript.
1. Abbreviations should be used only after defining them first. Example: FDA, CE, etc.
2. Format of Tables should be adjusted, e.g. Table 2, Table 3, etc.
3. Please check the manuscript for format, e.g. Figure 7, etc.
4. Better expression of the manuscript is expected.
I suggest to revise the manuscript to make it proper representative of the presented work.
Comments on the Quality of English LanguageExtensive editing of English language required.
Reviewer 2 Report
Comments and Suggestions for Authors
The measurement of heart rate and respiratory rate is a basic function for state--of-the-art PPG devices. There are some flaws in this study which warrant further improvement.
1. The sample size is very small. There is no justification based on sample size calculation.
2. The direct comparison of heart rate/respiratory rate can be misleading. It is recommended to compare the waveforms from the two sensors.
3. Spearman’s coefficient was focused in the abstract. Whereas, with wide ranges between LoAs, the performance is not comparable to some latest wearable sensors (mechanical ones and PPG). Especially, an error of respiratory rate higher than two breaths per minute could lead to misdiagnosis.
4. Finally, the innovation of this paper is questionable considering the existing miniaturized PPG sensors.
I encourage the authors to include more subjects and perform a comprehensive comparison on waveforms for more quantitative results.
Comments on the Quality of English LanguageOverall the language is OK but can be further improved.
Reviewer 3 Report
Comments and Suggestions for Authors
The topic of this paper regarding Measuring of Heart- and Respiratory Rate based on Off-the-shelf photoplethysmographic hard- and open-source software is interesting. The paper is generally well-written and seems to share comprehensive information on the topic with respect to the current literature. I have some remarks before I recommend the manuscript for publication:
Minor comments:
The authors should explain all abbreviations before their first use throughout the manuscript.
The authors should check the spelling.
What differences in measurement parameters did you find for different skin types?
How accurate was the placement of the electrodes?
How do you think sweat production can affect measurements?
Was the area under the electrode specially cleaned before the measurement or was used any special gel?
Comments on the Quality of English LanguageThe authors should check the spelling.
Reviewer 4 Report
Comments and Suggestions for Authors
This manuscript describes a system for recording a pulse wave using a circuit based on Arduino. The authors put forward a hypothesis about the possibility of using this scheme to record heart rate and respiratory rate.
Despite the high-quality work, it should be noted that the following points require mandatory improvement:
1. The novelty of the developed scheme is not obvious from the text. Schemes for connecting a photoplethyssographic sensor to an Arduino are widely described in scientific and popular sources.
2. The positioning of the sensor is questionable. Which artery is the sensor attached to? Has the method of attaching the sensor been worked out by ordinary users?
3. The article does not provide an example of a recorded signal
4. The article does not provide descriptions of the tables
5.Tables 3 and 4 require indication of dimensions
6. Figures 5-7 are not very informative.
7. Typo in line 219
Round 2
Reviewer 1 Report
Comments and Suggestions for Authors
In this manuscript, a guide to measure heart- and respiratory rate is presented. The study is interest to readers.
However, there are several small problems in the manuscript.
1. In Abstract, abbreviations should be used only after defining them first. Example: FDA, CE, etc.
2. Please check the manuscript for format.
I suggest to revise the manuscript before publishing.
Comments on the Quality of English Language
Minor editing of English language required
Author Response
Dear Reviewer,
First of all we would like to thank you for the remarks given. We have gone trough the manuscript an added the required ajustements.
- In Abstract, abbreviations should be used only after defining them first. Example: FDA, CE, etc.
- Abbreviations have been eliminated from the abstract.
- Please check the manuscript for format. I suggest to revise the manuscript before publishing.
- We went trough it we the entire team and adapted some sentences and ordering of the text. All changes are indicated in Red in the newly uploaded manuscript.
By this we hope to have satisfied your remarks.
Reviewer 4 Report
Comments and Suggestions for Authors
All comments have been corrected. The manuscript may be accepted
Author Response
Dear reviewer,
We would like to thank you for your insights and contribution to reviewing our manuscript.
Many thanks and with kind regards,
Guylian Stevens